# Effect of Aerobic Exercise Training on Sleep and Core Temperature in Middle-Aged Women with Chronic Insomnia: A Randomized Controlled Trial

**DOI:** 10.3390/ijerph20085452

**Published:** 2023-04-10

**Authors:** Pauline Baron, Éric Hermand, Valentin Bourlois, Thierry Pezé, Christophe Aron, Remi Lombard, Rémy Hurdiel

**Affiliations:** 1Univ. Littoral Côte d’Opale, Univ. Artois, Univ. Lille, ULR 7369—URePSSS—Unité de Recherche Pluridisciplinaire Sport Santé Société, F-59140 Dunkerque, France; 2Centre Sommeil Morphée, Polyclinique de Grande Sysnthe, 59760 Grande Synthe, France; 3Clinique de Flandre, 59210 Coudekerque-Branche, France

**Keywords:** sleep, exercise, health, core temperature

## Abstract

Background: Exercise represents a viable non-pharmacological intervention to help treating insomnia but the interaction mechanisms between sleep and physical activity still remain poorly understood. The aim of this study was to investigate the effect of a aerobic exercise training intervention on sleep and core temperature. Methods: Twenty-four adult women suffering from insomnia participated in this study. They were randomized into an exercise group and a control group. Aerobic exercise training consisted in moderate to vigorous aerobic exercise training for 12 weeks. Outcome measures included both subjective (Insomnia Severity Index, ISI) and objective (actigraphy recordings) sleep quality assessments, and core body temperature continuously recorded for a minimum 24 h. Results: The exercise group showed a decrease in ISI (*p* < 0.001) and in various objective sleep parameters. The core temperature batyphase value was lowered (*p* = 0.037) whereas its amplitude was larger (*p* = 0.002). We also found a tight correlation between the evolution of insomnia and the evolution of mean night-time core temperature and batyphase values. Conclusions: A moderate to vigorous aerobic exercise program appears to be an effective non-drug therapy for improving sleep in women with insomnia. In addition, exercise programs should aim to increase core body temperature during practice to induce sleep-promoting adaptations and rebound.

## 1. Introduction

Chronic insomnia is a sleep disorder characterized by persistent difficulties in initiating and maintaining sleep, early awakenings and non-restorative sleep, leading to alterations during the day such as fatigue, altered mood, concentration problems and poorer quality of life [1]. The hyperactivation theory suggests that insomnia is associated with inappropriate physiological arousals [2]. In addition, Lack et al. [3] provided a review of the literature regarding insomnia and showed that an elevated core temperature is a marker of physiological hyperexcitation.

Drug therapy is currently the most prescribed treatment for insomnia. However, sleeping pills may cause deleterious side effects and are not recommended for a long-term use. Thus, various non-pharmacological therapies have been developed, including cognitive and behavioral therapies [4]. However, the main disadvantage of this approach is the lack of wide availability because it must be done by highly trained specialists [5]. Hence, exercise may be an interesting long-term treatment, inexpensive and easily accessible to prevent and treat insomnia [6,7]. A recent meta-analysis showed a positive effect of physical activity on insomnia symptoms [8]. However, the authors did not show improvement in sleep measured by actigraphy or polysomnography. However, further studies are needed to confirm these findings.

The mechanisms by which physical activity might reduce the symptoms of insomnia are not yet well identified [9]. Several hypotheses have been put forward. For example, exercise may improve mood, which may be an important factor in decreasing insomnia symptoms [10]. Also, it has been suggested that increased energy expenditure and depletion of glycogen reserve during exercise would lead to a need for sleep: this is the theory of body restoration and energy conservation [11]. Sleep is also characterized by important changes in the circadian rhythm of heart rate, blood pressure and core temperature induced by declining sympathetic and increasing parasympathetic activity during the night [12,13]. The beneficial effects of exercise on insomnia could also originate from its effects on the autonomic nervous system (ANS) by increasing parasympathetic and decreasing sympathetic activities [14,15]. The majority of studies relied on heart rate variability (HRV) as a tool to assess ANS activity. The measurement of core temperature, as a reflection of ANS activity, may also be of interest in view of the very close link between core temperature and insomnia [16]. To our knowledge, no study has examined the effect of an exercise program on nocturnal core temperature in people with insomnia.

The first aim of this study was to investigate the effect of a 12-week aerobic exercise training intervention on insomnia severity and objective sleep in adult women suffering from insomnia. Indeed, the effect of a physical activity program on insomnia symptoms is well documented, and articles point to a decrease in insomnia symptoms through exercise. However, the results on the effect of a physical activity program on objective sleep parameters in insomniacs are less conclusive. Thus, our study brings new results on this debate. The second objective was to identify some potential interaction mechanisms of the association between insomnia and exercise with a special focus on the core temperature. Knowledge of these interaction mechanisms could help to identify the optimal exercise modalities to reduce insomnia and allow the optimization of PA programs by inducing identified and specific adaptations.

## 2. Materials and Methods

### 2.1. Study Design

This randomized controlled trial divided inactive and insomniac women aged 18 to 64 years into two groups: Exercise program (E) or control (C) in a 1:1 ratio. This study was approved by the committee for the protection of individuals.

This study was conducted from September 2021 to December 2022 in Dunkirk area.

### 2.2. Participants

Inclusion criteria were that participants were: (1) women, (2) aged 18–64 years, (3) meeting the DSM-5 diagnostic criteria for chronic insomnia and (4) being inactive, i.e., not meeting the World Health Organization recommendations for physical activity in the past 6 months. Exclusion criteria were: (1) any acute illness that might interfere with the study, (2) presence of a major psychiatric disorder, (3) high risk of sleep apnea (assessed by the Berlin questionnaire), (4) symptoms of restless legs syndrome (assessed by the International Restless Legs Syndrome Study Group rating scale), (5) alcohol consumption greater than 3 drinks per day, (6) working shifts or other imposed irregular work schedules, (7) having experienced jet lag in the previous month, (8) having a body mass index (BMI) > 35 kg/m^2^, and/or (9) refusing to adhere to the study protocol.

### 2.3. Procedures

Volunteering participants underwent an assessment phase before inclusion. An interview conducted by the principal investigator and the administration of various questionnaires ensured that participants met the eligibility criteria.

### 2.4. Randomizations and Interventions

After giving written informed consent, participants who met the eligibility criteria were matched in pairs according to age: one participant was assigned to the exercise group, the other to the group. Randomization was performed using a computer-generated program independent of treatment staff. A total of 24 participants were enrolled and randomly assigned to the E or C groups.

### 2.5. Intervention Group

The exercise sessions were divided into 3 weekly sessions, each of them lasting 1 h and 15 min (225 min per week). Active walking was performed outdoors if the weather allowed it or on an indoor treadmill. An experienced trainer was specifically assigned to each single participant so it was possible to accurately adjust the session intensity. Indeed, each participant wore a heart rate (HR) sensor (OH1) (Polar Electro Oy, Kempele, Finland) [17] connected to the trainer’s smartphone and was regularly asked for her rating of perception exertion (RPE) (BORG CR-10). The maximal HR was estimated by the Astrand and Rhyming equation (220-age). During each session, the objective was to spend 50 min in moderate intensity (70–80% HR max; RPE: 4–6) and 25 min in vigorous intensity (>80% HR max; RPE > 6). On a weekly basis, each participant spent approximately 165 min in moderate intensity and 75 min in vigorous intensity, which largely met the World Health Organization recommendations.

### 2.6. Control Group

Participants were asked to continue their usual lifestyle, and to maintain their baseline physical activity levels.

### 2.7. Data Collected

Assessments were performed at baseline (T0), i.e., before the intervention, and at 12 weeks, i.e., after the intervention (T1).

### 2.8. Body Composition

Body mass index (BMI) and percentage of fat mass were measured by an impedance scale (Tanita DC-360) (Tanita Corp., Tokyo, Japan).

### 2.9. Sleep

Symptoms of insomnia was measured by the Insomnia Severity Index (ISI) [18]. The total score ranged from 0 to 28, a score of 15 or higher being considered as clinical insomnia.

Sleepiness was measured by the EPWORTH scale [19]. A score greater than or equal to 9 reflects high daytime sleepiness.

Objective sleep was measured by an accelerometer (GT3X, TheActiGraph, Pensacola, FL, USA) over 7 nights [20]. Participants wore the accelerometer on the wrist of their non-dominant arm. They also completed a daily sleep diary: wake-up and bedtime times, naps. The accelerometer data were retrieved and analyzed using Actilife 6.0^®^ software, using the Cole-Kripke algorithm [21]. All epochs (one given every 60 s) were automatically classified as sleep or wake based on the total number of activities for the given epoch and a predefined wake/sleep threshold value. Bedtime and wake-up times were defined and applied using the subjects’ sleep diaries.

Sleep quantity parameter was:-Total Sleep Time (TST, min): duration between falling asleep and waking up, not including wakefulness periods.

Sleep quality parameters were:-Sleep Efficiency (SE, %): ratio of total sleep duration to duration in bed;-Wake After Sleep Onset (WASO, min): total time awake after sleep onset. For a better understanding, WASO was expressed as a function of TST: WASO/TST × 100 (%);-Sleep Latency (SL, min): refers to the duration for a subject to fall asleep, between the time of bedtime, reported by the participant, and the moment when the subject was asleep, reported by the algorithm.

### 2.10. Core Temperature

The “e-celsius bodycap” capsule was used to measure the patient’s core temperature continuously for at least 24 h. Although no individual calibration of pills was performed, previous research has shown the equipment to be highly accurate and reliable [22].

The time of intake was standardized for all participants at 3 p.m. Participants should not perform any intense exercise at least 4 h before bedtime. The temperature was recorded at 30 s intervals. The data were analyzed by the e performance manager software. The values of batyphase (minimum temperature) and acrophase (maximum temperature) over a 24-h period (3 p.m.-3 p.m.), the amplitude (temperature difference between acrophase and batyphase) and the average temperature over the time spent in bed were recovered for analysis.

### 2.11. Cardiorespiratory Fitness

YMCA incremental submaximal test on an ergometer, up to 85% of theoretical HR max, was used to evaluate the Maximum Aerobic Power (MAP, W) and the associated $\dot{V}$O_2_max (mL·min^−1^·kg^−1^) [23].

### 2.12. Stress, Anxiety and Depression

The Hospital Anxiety and Depression scale (HAD) was also administered. A score greater than or equal to 11 reflects anxious and depressive symptomatology [24].

Stress was measured by the Perceived Stress Scale (PSS-10). A score of 10 to 50 is given, a score above 27 reflects high perceived stress [25].

### 2.13. Statistical Analysis

Results are presented as mean ± standard deviation (SD). Normality and variance homogeneity of data were verified by Shapiro-Wilk and a Levene tests, respectively.

Tests of comparisons at T0 and T1 between the two groups were performed by the Student t test. Cohen’s effect size (d) was calculated by subtracting the final value from the initial value and dividing by the pooled standard deviation. According to convention, effect sizes of 0.2–0.3, 0.5, and >0.8 are considered small, medium and large, respectively. The primary question was focused on the benefits of exercise program compared with the control group in adult women with insomnia. For each condition (C and E), linear mixed models with time as a fixed effect and subject as a random effect were performed to determine whether the within-person values differed between T0 and T1. These models were adjusted for age. Then, to compare the differences in changes between the two groups, student t tests were performed.

Pearson correlations were then performed to assess whether changes in the primary variables were associated with changes in the measurements of the secondary variables. Analyses were performed with R software. Statistical significance was defined as *p* < 0.05.

### 2.14. Sample Size Calculation

Sample size estimation was performed using R. The effect size for a randomized controlled trial to evaluate the effect of an exercise program on sleep quality in individuals with insomnia was set at 0.87 [6]. By setting beta at 20% and alpha at 5%, a ratio of 1:1, it was found that a minimum of 12 participants per group was needed to detect this effect size in the primary outcome (ISI).

## 3. Results

Concerning the compliance rate, the 24-people included (12 in each group) were at the end of the study. In the exercise group, out of the 36 sessions performed by one person, on average 2 ± 1 sessions (5.6%) were cancelled, mostly for medical reasons.

At T0, all variables were not different between the Control Group and the Exercise Group (Table 1).

At T1, the Exercise group had a lower ISI score and EPWORTH score (*p* < 0.001; d = 3.00 and *p* = 0.026; d = 0.97, respectively) and higher core temperature amplitude (*p* = 0.007; d = −1.04) than the Control group (Table 1).

In the exercise group, there was an average 8-point reduction in the ISI score and 5-point reduction in the EPWORTH score (Table 2). For objective sleep variables, in the exercise group, there was an average 4% increase in sleep efficiency, 33 min increase in total sleep time and 4% decrease in wake after sleep onset (Table 2). Concerning core temperature, the exercise group increased its amplitude and decreased its batyphase value (Table 2). Finally, the exercise group also increased its VO_2_max and decreased its PSS-10, and HAD scores (Table 2).

While for the control group, there was a reduction in sleep latency and in HAD score (Table 2).

The changes between the two group were different for ISI score (*p* < 0.001; d = 2.63), EPWORTH score (*p* = 0.021; d = 0.713), TST (*p* = 0.033; d = −0.729), WASO (*p* = 0.011; d = 0.682), PSS-10 score (*p* = 0.029; d = 0.860), VO_2_max (*p* = 0.022; d = −1.36), nocturnal mean temperature (*p* = 0.032; d = 1.07), batyphase value (*p* = 0.048; d = 1.01) and amplitude (*p* = 0.37; d = −1.22) (Table 2).

Upon comparing the change of the number of participants in the two categories of ISI (<15 vs. ≥15) within the groups over time, there was a reduction of 83% (10/12 participants) in the clinical insomnia category of the intervention group, compared with 8% (1/12 participants) reduction within the same category in the control group (Figure 1).

Changes in ISI score were correlated with changes in batyphase value (r = 0.691, *p* < 0.001; Figure 2), in mean night-time temperature (r = 0.704, *p* < 0.001), and in core temperature amplitude (r = −0.688; *p* < 0.001). Changes in objective sleep parameters were not correlated with changes in core temperature variables.

Finally, changes in ISI score were also correlated with changes in EPWORTH score (r = 0.614, *p* = 0.003).

## 4. Discussion

The first objective of the study was to investigate the effect of a 12-week aerobic exercise training intervention on insomnia severity in adult women suffering from insomnia. The result of our study showed that a moderate to vigorous intensity aerobic exercise program is effective in reducing the severity of insomnia and daytime sleepiness in middle-aged women. Following an exercise program also increased their sleep efficiency and total sleep time, and decreased their wake after sleep onset.

The results of this study are therefore consistent with the existing literature about the benefits of exercise on insomnia severity [26,27,28]. In the study by Hartescu et al. [29], after a 6-month moderate aerobic exercise program, participants decreased their ISI score by 4 points and 67% of participants in the PA group moved from clinical to subclinical insomnia. Although these data are in line with the present study, we found a greater decrease in ISI score in the exercise group. We may bring two hypotheses to explain the observed differences. First, the intensity of their exercise program was moderate, and vigorous exercise has been shown to decrease insomnia symptoms more than moderate [30]. Second, a quarter of the participants in Hartescu’s study were men: sex may modulate the effectiveness of an exercise program and women might be more receptive to exercise, especially at vigorous intensity [27].

In our study, the reduction in the severity of insomnia is also associated with a similar evolution of the objective data, illustrated by an increase in total sleep time and sleep efficiency, and a decrease in wake after sleep onset. Results in the literature are more mixed on whether exercise has an impact on objective sleep. A previous meta-analysis showed small beneficial effect of an exercise program on total sleep time, sleep efficiency and sleep latency [27]. However, our data did not indicate any effect on sleep latency. This can be explained by the fact that our participants did not exhibit problems of falling asleep before the program (SL = 7 min). A more recent meta-analysis [7] highlighted a mild decrease of wake after sleep onset in the exercise group compared, which is in accordance with our study. However, the last meta-analyses cited included studies with participants who were not necessarily insomniacs. A very recent meta-analysis analyzed the effect of a PA program in people with insomnia [8]. As in our study, they showed a reduction in subjective complaints but did not find an effect on objective sleep. However, only one study of the six included had used an actimeter (the others used polysomnography). In this study using actimeter, they also used polysomnography [31]. They showed that a PA program had an effect on SL, WASO, and SE measured by actimeter, but they did not find these results on these same parameters but measured by polysomnography.

Along with this decrease of insomnia symptoms and improvement of objective sleep, we observed a decrease of daytime sleepiness. Furthermore, the evolution in EPWORTH score was correlated to the evolution in ISI score indicating that changes in insomnia severity may have had a positive impact on sleepiness.

The exercise group also significantly increased their VO_2_max. In addition to the overall health benefits, increasing VO2max in middle-aged individuals with insomnia may bring further advantages: some studies indeed showed a protective effect of cardiorespiratory fitness against sleep disorders [32].

The second objective was to identify some potential interaction mechanisms of the association between insomnia and exercise. Among the different potential mechanisms of interaction between insomnia and exercise, our study was focused in the modification of stress [33], in anxiety and depressive symptoms [34] as well as the modification of core body temperature at night [3]. In our study, following an exercise program decreased their perceived stress, and anxiety and depressive disorders. Regarding the core temperature, we observed a decrease of batyphase value while the amplitude over 24 h increased.

In the present study, the exercise intervention reduced perceived stress, depression and anxiety (i.e., PSS and HAD scores), which is accordance with the existing literature [10,29].

The exercise program also decreased the batyphase value and increased the 24-h core temperature amplitude. In a study by Horne and Moore [35], participants performed, under two conditions, a treadmill run at 75% of their VO_2_max for 2 × 40 min between 14:30 and 17:30. The first condition included additional clothing inducing an increase in rectal temperature of 2.3 °C during exercise; the second condition, with additional cooling, inducing an only increase of 1 °C during exercise. Only the first condition had a positive impact on sleep with an increase in deep slow wave sleep time. The increase in core temperature induced by acute exercise could therefore activate thermolysis mechanisms at the end of it in order to cool the body by rebound effect and thus decrease night-time core temperature and decrease insomnia.

From a chronic point of view, physical activity could regulate the activity of the autonomic nervous system (ANS). Indeed, a recent meta-analysis showed a slight increase in RMSSD and HF after a PA program reflecting an increase in parasympathetic modulation [36].

The sympathetic axis of the ANS is responsible for vasoconstriction of the smooth muscles surrounding the arteriovenous anastomoses, resulting in a decrease in distal skin temperatures and thus an increase in core temperature [37]. Thus, the decrease in nocturnal core temperature could reflect a decrease in sympathetic and/or an increase in parasympathetic modulation of the ANS.

The evolution in nighttime temperature and batyphase is tightly correlated with the evolution of insomnia severity. Thus, strategies to improve thermoregulation and activate thermolysis mechanisms during sleep seem to be relevant in the management of insomnia.

Our study has some limitations. For organizational reasons, the time of exercise session presented inter- and intra-individual differences. Because exercise was performed outdoors, depending on the time of day and season in which the program took place, light levels and environmental temperature could not be controlled. Given the variation in body temperature and sleep metabolic rate during the menstrual cycle [38], controlled studies about the effects of exercise in the follicular and luteal phases are needed. In addition, the age range of our study being wide, we probably included non-menopausal women and menopausal women. This could lead to bias and errors in the insomnia evaluation We are also well aware that core temperature can be influenced by other factors. Electroencephalogram sleep measurement would provide data on the effect of the exercise program on sleep stages. Finally, because the participants were all women, the findings may not be generalizable to the entire population.

## 5. Conclusions

In conclusion, our results underline that aerobic physical training performed at moderate to vigorous intensity has a positive effect on insomnia severity and nocturnal core temperature. Furthermore, we found that the evolution of insomnia can be strongly associated with the evolution of nocturnal core temperature. Our results suggest that exercise interventions should be built to induce a rise in body temperature in order to induce sleep-promoting adaptations and rebounds. Further studies with larger sample sizes are needed to confirm these results.

## Figures and Tables

**Figure 1 ijerph-20-05452-f001:**
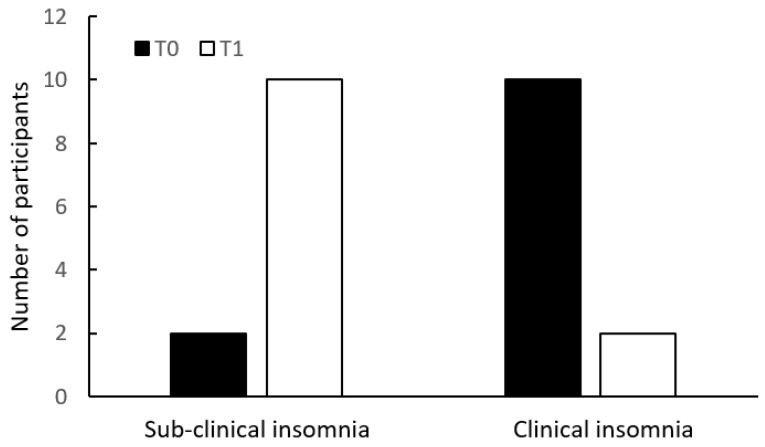
Insomnia Severity Index (ISI) categories at T0 and T1, in the intervention group (**above**) and in the control group (**below**).

**Figure 2 ijerph-20-05452-f002:**
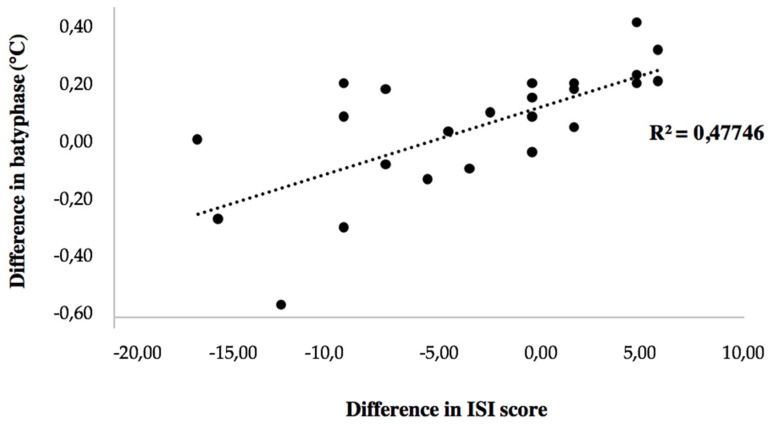
Correlation between change in batyphase and change in ISI score.

**Table 1 ijerph-20-05452-t001:** Variables at T0 and T1 for each group. * *p* < 0.05 from the control group at T1. ** *p* < 0.01 from the control group at T1. *** *p* < 0.001 from the control group at T1.

	T0		T1	
	Exercise Group	Control Group	Exercise Group	Control Group
Age (years)	46.4 ± 5.67	44.8 ± 7.82	46.6 ± 5.67	44.8 ± 7.82
BMI (kg·m^−2^)	24.9 ± 4.69	23.8 ± 2.34	24.8 ± 3.98	24.0 ± 2.55
Fat mass (%)	29.1 ± 6.76	27.4 ± 4.15	28.4 ± 5.81	27.6 ± 4.80
Epworth	11.8 ± 7.47	13.3 ± 3.63	6.97 ± 4.36 *	12.6 ± 6.92
ISI	17.1 ± 3.32	17.5 ± 2.11	8.93 ± 3.63 ***	18.5 ± 2.67
TST (min)	432 ± 80.5	435 ± 29.5	465 ± 68.5	442 ± 26.2
WASO (%)	16.7 ± 11.6	16.5 ± 10.0	12.6 ± 9.03	15.86 ± 8.99
SE (%)	79.8 ± 8.01	81.1 ± 7.59	83.6 ± 6.43	82.13 ± 6.09
SL (min)	6.98 ± 4.29	7.72 ± 3.91	6.26 ± 4.74	3.91 ± 1.53
PSS-10	30.1 ± 5.52	28.3 ± 5.48	25.4 ± 4.44	26.9 ± 4.29
HAD	17.3 ± 6.20	17.8 ± 3.07	14.6 ± 5.76	16.6 ± 3.37
$\dot{V}$O_2_max (mL·min^−1^·kg^−1^)	29.4 ± 4.34	31.7 ± 6.02	31.0 ±4.46	31.5 ± 5.63
Nocturnal mean temperature (°C)	36.8 ± 0.15	36.8 ± 0.15	36.7 ± 0.19	36.9 ± 0.23
Batyphase (°C)	36.5 ± 0.20	36.5 ± 0.25	36.3 ± 0.22	36.5 ± 0.315
Acrophase (°C)	37.6 ± 0.23	37.7 ± 0.27	37.7 ± 0.20	37.6 ± 0.07
Amplitude (°C)	1.15 ± 0.29	1.23 ± 0.54	1.40 ± 0.24 **	1.13 ± 0.28

**Table 2 ijerph-20-05452-t002:** Standardized β-Coefficient (95% confidence interval) of changes in variables by group and differences in changes between the two group.

	Exercise Group		Control Group			
	Standardized β-Coefficient (95% Confidence Interval)	*p*	Standardized β-Coefficient (95% Confidence Interval)	*p*	Net Effect	*p*
ISI	−8.17 (−10.8; −5.54)	**<0.001**	0.95 (−1.29; 3.21)	0.102	−9.12	**<0.001**
SE (%)	3.84 (1.12–6.56)	**0.015**	1.03 (−0.04; 2.10)	0.075	2.81	0.222
SL (min)	−0.72 (−2.46; 1.02)	0.416	−3.81 (−7.06; −0.56)	**0.036**	−3.09	0.152
TST (min)	32.8 (11.1; 54.5)	**0.011**	7.12 (−5.25; 19.5)	0.266	25.7	**0.033**
WASO (%)	−4.12 (−7.59; −0.65)	**0.034**	−0.64 (−1.68; 0.40)	0.234	−3.48	**0.011**
Epworth	−4.83 (−8.68; −0.99)	**0.026**	−0.67 (−2.00; −0.16)	0.198	−4.16	**0.021**
BMI (kg/m^2^)	−0.13 (−0.33; 0.59)	0.570	0.22 (0.12; 0.94)	0.619	−0.35	0.518
Fat mass (%)	−0.68 (−1.47; 0.12)	0.111	0.17 (0.01; 0.31)	0.405	−0.85	0.196
PSS-10	−4.67 (−7.10; −2.23)	**0.003**	−1.41 (−2.10; −0.35)	0.315	−3.26	**0.029**
HAD	−2.67 (−5.08; −0.26)	**0.046**	−1.25 (−2.05; −0.45)	**0.009**	−1.42	0.396
$\dot{V}$O_2_max (mL·min^−1^·kg^−1^)	1.55 (0.73; 2.37)	**0.003**	−0.20 (−0.47;0.07)	0.166	1.75	**0.022**
Nocturnal mean temperature (°C)	−0.12 (−0.25; −0.01)	0.096	0.10 (0.04; 0.16)	0.078	−0.22	**0.032**
Batyphase (°C)	−0.16 (−0.29; −0.02)	**0.037**	0.04 (0.01; 0.09)	0.063	−0.20	**0.048**
Acrophase (°C)	0.10 (0.01; 0.21)	0.084	−0.06 (−0.10; 0.13)	0.496	0.16	0.413
Amplitude (°C)	0.26 (0.13; 0.38)	**0.002**	−0.10 (−0.28; 0.08)	0.280	0.36	**0.037**

Note: *p* values (<0.05) are in bold.

## Data Availability

Data available on request due to restrictions The data presented in this study are available on request from the corresponding author. The data are not publicly available due to privacy.

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
