# Peer review of "Effect of Aerobic Exercise Training on Sleep and Core Temperature in Middle-Aged Women with Chronic Insomnia: A Randomized Controlled Trial"

_ijerph, 2023, doi:10.3390/ijerph20085452_

Round 1

Reviewer 1 Report

This is a very interesting RCT about the effect of physical exercise on sleep and core temperature. I have a few comments:

1. To be more consistent with the authors’ interpretation, it makes more sense to present Figure 1 differently. The top panel can be about the intervention group, and the bottom panel about the control group. Within each panel for each indicator, put two bars side by side: one for T0, and the other for T1. In this way, changes between T1 and T0 would be clearer for readers.

2. Lines 159: linear mixed models evaluated the change in each variable with time as a fixed factor and subject as a random factor. This description is not quite clear to me. Can the authors elaborate a little?

3. In an experiment design, you compare changes between T1 and T0 in the intervention group and control group. However, Table 2 and the results section only evaluate changes in each group without computing the differences in changes. For example, for ISI, the change in the intervention group is -8.17, and that in the control group is 0.95. The net effect would be -9.12. So there should be another two columns for the net effects and p values.

4. Line 278. Another limitation is that because the participants were all women, the findings may not be generalizable to the entire population.

Author Response

Reviewer 1:

  1. To be more consistent with the authors’ interpretation, it makes more sense to present Figure 1 differently. The top panel can be about the intervention group, and the bottom panel about the control group. Within each panel for each indicator, put two bars side by side: one for T0, and the other for T1. In this way, changes between T1 and T0 would be clearer for readers.

Author’s response: We took your comments into account and modified Figure 1 as you suggested.

  1. Lines 159: linear mixed models evaluated the change in each variable with time as a fixed factor and subject as a random factor. This description is not quite clear to me. Can the authors elaborate a little?

Author’s response: We rephrased the sentence for a better understanding. (Page 4, line 180).

“linear mixed models with time as a fixed effect and subject as a random effect were performed to determine whether the within-person values differed between T0 and T1.”

  1. In an experiment design, you compare changes between T1 and T0 in the intervention group and control group. However, Table 2 and the results section only evaluate changes in each group without computing the differences in changes. For example, for ISI, the change in the intervention group is -8.17, and that in the control group is 0.95. The net effect would be -9.12. So there should be another two columns for the net effects and p values.

Author’s response: Thank you for your relevant comment. We added the two columns in table 2 as suggested. We added a sentence to explain what we did in the statistical analysis part: “Then, to compare the differences in changes between the two groups, student t tests were performed.”

We have also added a sentence to explain these results in the results section: “The changes between the two group were different for ISI score, EPWORTH score, TST, WASO, PSS-10 score, VO2max, nocturnal mean temperature, batyphase value and amplitude (Table 2).”

  1. Line 278. Another limitation is that because the participants were all women, the findings may not be generalizable to the entire population.

Author’s response: indeed, we have added this limitation at the end of the manuscript. (Page 11, line 362)

Reviewer 2 Report

Dear authors,

I carefully read your manuscript aiming to assess the effects of 4-week aerobic training on insomnia and temperature in a sample of 24 women.

Please, find my comments below:

INTRODUCTION:

1.       It does not cover the preexisting literature or shed light on what new your research is bringing. Moreover, physical activity does not only act on the nervous system in modifying sleep quality. I suggest deepening the literature by reading : doi: 10.31744/einstein_journal/2022AO8058, https://doi.org/10.1093/ajh/hpab146, DOI: 10.5603/GP.a2020.0172, DOI: 10.4103/jmh.JMH_35_19, doi: 10.1053/smrv.2000.0110, doi: 10.1038/s41598-022-25014-7.

2.       You speak about insomnia; however, your aim and most of your results focus on sleep rather than insomnia.

METHODS

3.       The age range is vast; you could have involved menopausal women with different sleep problems from premenopausal women. This could lead to bias and errors in the insomnia evaluation

4.       It would be best to specify everywhere (title, introduction, methods, etc.) that your exercise intervention is anaerobic.

5.       The citation is missing in the sleep parameters explanation; I suggest adding doi: 10.1080/07420528.2022.2157737.

6.       It is difficult to understand the circadian rhythm parameters of body temperature without an exhaustive.

7.       The statistical analysis does not reflect the results: where did you report the linear mixed model analysis?

RESULTS

8.       Data at T1 are not reported, and it is insufficient to give only the significant p-values in lines 178-180.

9.       Figure 1 does not represent what is explained in lines 186-189.

10.   In general, the result section needs to be better organized, the data need to be deeper exposed, and the results better explained (for example, what do the results in lines 205-208 mean?).

11.   Do the temperature parameters correlate with the objective sleep parameters? In the last part of your results, which is also your key result, you correlate objective and subjective data; it is not a robust and reliable outcome.

DISCUSSION

12.   It needs to be reviewed in light of the added papers suggested for the introduction and to better explain the results obtained. Please, keep in mind that the action of physical activity on sleep is not merely related to ANS modulation.

Author Response

INTRODUCTION:

  1. It does not cover the preexisting literature or shed light on what new your research is bringing. Moreover, physical activity does not only act on the nervous system in modifying sleep quality. I suggest deepening the literature by reading : doi: 10.31744/einstein_journal/2022AO8058, https://doi.org/10.1093/ajh/hpab146, DOI: 10.5603/GP.a2020.0172, DOI: 10.4103/jmh.JMH_35_19, doi: 10.1053/smrv.2000.0110, doi: 10.1038/s41598-022-25014-7.

Author’s response: Thank you for these very interesting references. We modified and developed the introduction. We emphasized the fact that other mechanisms could improve sleep through exercise.

  1. You speak about insomnia; however, your aim and most of your results focus on sleep rather than insomnia.

Author’s response: Thank you for your comment. We focused our work on insomnia because the people recruited suffered from insomnia according to the DSM-5 criteria and the questionnaire used to assess sleep was the Insomnia Severity Index (and not the Pittsburgh Sleep Quality Index for example).

Also, the theory of hyperarousal, which may be reflected by a nocturnal core temperature, is specific to insomnia (doi: 10.1016/j.smrv.2008.02.003. ;  doi: 10.1016/j.smrv.2009.04.002.)  

METHODS

  1. The age range is vast; you could have involved menopausal women with different sleep problems from premenopausal women. This could lead to bias and errors in the insomnia evaluation

Author’s response: Indeed, we are aware of this limitation. Age being a covariate in the linear mixed model, it allows us to limit this bias. However, we added the following sentence in the limitations of the study (Page 11, line 358).

“In addition, the age range of our study being wide, we probably included non-menopausal women and menopausal women. This could lead to bias and errors in the insomnia evaluation.”

  1. It would be best to specify everywhere (title, introduction, methods, etc.) that your exercise intervention is anaerobic.

Author’s response: The American College of Sport Medicine defines aerobic exercise as any activity that involves large muscle groups, which can be maintained continuously. Anaerobic exercise has been defined by the ACSM as intense physical activity of very short duration and independent of inhaled oxygen use.

In our study, a physical activity session lasted 1 hour and 15 minutes with 50 min of moderate intensity and 25 min of vigorous intensity (walking or running). Although we proposed intense physical activity, the session was continuous and the majority of our participants used their aerobic pathway. Moreover, the criterion for measuring aerobic capacity is maximum oxygen consumption (VO2). The fact that we observed an increase in VO2max in the physical activity group supports the idea that the proposed sessions were predominantly aerobic.

However, we are listening to your point of view and we would be interested to understand why you would define the proposed exercise as anaerobic.

  1. The citation is missing in the sleep parameters explanation; I suggest adding doi: 10.1080/07420528.2022.2157737.

Author’s response: thank you, we have added this citation. (Page 3, line 130).

  1. It is difficult to understand the circadian rhythm parameters of body temperature without an exhaustive.

Author’s response: Core temperature may indeed be a reflection of circadian rhythm (doi: 10.1016/j.ncl.2019.05.001). However, in our study, we used core temperature as a reflection of autonomic nervous system activity (doi: 10.1016/j.neulet.2018.11.027). We are also well aware that core temperature can be influenced by other factors. We have added this limitation at the end of the manuscript (Page 11, line 359).

  1. The statistical analysis does not reflect the results: where did you report the linear mixed model analysis?

 Author’s response: The results of the linear mixed model are presented in Table 2 with Standardized β-Coefficient and p value. We added the 95% confidence interval, as did this study for example (Table 3: doi: 10.3390/ijerph18157999).

RESULTS

  1. Data at T1 are not reported, and it is insufficient to give only the significant p-values in lines 178-180.

Author’s response: We reported all data at T1 in Table 1. We added the effect sizes.

  1. Figure 1 does not represent what is explained in lines 186-189.

Author’s response: We took your comments into account and modified Figure 1 accordingly for a better understanding.

  1. In general, the result section needs to be better organized, the data need to be deeper exposed, and the results better explained (for example, what do the results in lines 205-208 mean?).

Author’s response: We reviewed and detailed all our result section. We hope you will find it at your convenience.

  1. Do the temperature parameters correlate with the objective sleep parameters? In the last part of your results, which is also your key result, you correlate objective and subjective data; it is not a robust and reliable outcome.

Author’s response: Objective sleep parameters were not correlated with temperature variables. We specified this in the result section (page 8, line 255).

We are aware that correlating objective and subjective data is not optimal. However, in insomnia, it is also very important to take into account the person's perception. Moreover, polysomnography is not routinely recommended, except to rule out other sleep disorders. Indeed, patients suffering from insomnia tend to underestimate their total sleep time and overestimate their sleep onset latency and wake after sleep onset (doi: 10.1111/jsr.12046. Epub 2013 Mar 25). therefore, for the majority of people suffering from insomnia, the primary objective is to improve their perception of sleep.

DISCUSSION

  1. It needs to be reviewed in light of the added papers suggested for the introduction and to better explain the results obtained. Please, keep in mind that the action of physical activity on sleep is not merely related to ANS modulation.

Author’s response: We took into account your remark and revised the whole discussion section. We hope that this will suit you.

Reviewer 3 Report

This group randomized study tested the effect of aerobic exercise training on sleep and core body temperature in a sample (n=24) of women with insomnia. Participants were allocated to exercise or control groups. The duration of the follow-up was 12 weeks. Sleep was measured by both subjective and objective (accelerometry) methods. Sleep improvements were noted in the exercise group along with an increase in the amplitude of core body temperature. The authors suggest that exercise could be an effective treatment for sleep in this population.

The question is of interest. However, significant points need to be addressed and potential confounders are not even mentioned in the paper. Moreover, the presentation and interpretation of the results need extensive improvements. As it stands I do not recommend the publication of this paper in IJERPH.

Although the list is not exhaustive, please see below for reasons.

Major points:

-Further information should be given regarding the exercise intervention. For instance, it is not clear whether the exercise was realized in the morning, afternoon, or evening. Was the exercise of the same intensity for all participants and between sessions? How was that verified? How about the compliance rate throughout the study? More information regarding this point should be added. All these confounders may bias the results while no information was provided…

-Statistical analysis: The normality, homogeneity, and sphericity were verified for all analyses? Could you please report these data in your response? Please amend the figure (you should at least add the ecartype barre, What do you mean by AP?, Please added the p-value of the comparisons in table 1 (the title of table 1 should proceed the table). Effect sizes are recommended for all analyses. Table 2 can be done with the gain of each group, subtracting the posttest from the pretest, and performing an ANCOVA, having the pretest as a covariate. The unit should be added to Figure 2.

-Results section: could you further interpret the results?

Specific point

-Line 131: Please include the Epworth scale with the other subjective sleep measurements.

-Line 135-140: please built upon this paragraph. Further information on the analysis of core body temperature by the e-celsius bodycap should be added. For instance how the Batyphase (°C), Acrophase (°C), and Amplitude (°C) were calculated?

-The title of table 2 is not appropriate

- The conclusions must be humble, based on the sample size, with suggestions.

Author Response

Major points:

-Further information should be given regarding the exercise intervention. For instance, it is not clear whether the exercise was realized in the morning, afternoon, or evening. Was the exercise of the same intensity for all participants and between sessions? How was that verified? How about the compliance rate throughout the study? More information regarding this point should be added. All these confounders may bias the results while no information was provided…

Author’s response: Thank you for your comments.

As specified in the limitations section at the end of the discussion (page 10, line 353), practice time could not be controlled and presented intra and inter individual differences. To be more accurate, each person had the same schedule each week for 12 weeks but it could vary within the week (e.g., Monday: 12:30-1:45 pm; Wednesday: 5:30-6:45 pm; Friday: 5-6:15 pm).

As the sessions were individual, the hours of practice between two people could vary so that the supervisor could supervise several people in the same day. 

The relative intensity of the sessions was strictly the same for all participants and between sessions. During the session, participants were equipped with a Polar OH1 to measure their heart rate and were regularly asked for their Rate of Perceived Exertion. Everyone in each session spent 50 min in moderate intensity (70-80% HR max; RPE: 4-6) and 25 min in vigorous intensity (>80% HR max; RPE >6).

However, the absolute intensity (walking speed, % slope...) could vary between individuals and could also change over the 12 weeks. For example, for a participant, at the beginning of the program, walking at 5.5km/h on flat ground corresponded to an intense PA, then as the program progressed, her physical condition improved and to obtain an intense exercise, we had to walk faster and/or on hilly terrain.

Concerning the compliance rate, the 24-people included (12 in each group) were at the end of the study.  In the exercise group, out of the 36 sessions performed by one person, on average 2 ± 1 sessions (5.6%) were cancelled, mostly for medical reasons. We added this sentence, at the beginning of the result section.  

-Statistical analysis: The normality, homogeneity, and sphericity were verified for all analyses? Could you please report these data in your response?

Author’s response: For all the tests the normality of the data and the homogeneity of the variances were verified. For the linear mixed models used in our study, we checked:

  • The explanatory variables are related linearly to the response.
  • The errors have constant variance.
  • The errors are independent.
  • The errors are Normally distributed.

(Ref: Venables, William N., and Brian D. Ripley. 2002. Modern Applied Statistics with S-PLUS. New York: Springer.)

Please amend the figure (you should at least add the ecartype barre, What do you mean by AP?,

Author’s response: We took your comments into account and presented Figure 1 differently. We hope it is clearer for you. We did not add a standard-error bar as it is a number of participants (a distribution and not an average).

Effect sizes are recommended for all analyses.

Author’s response: Thanks for your comment, we added the effect sizes.

Table 2 can be done with the gain of each group, subtracting the posttest from the pretest, and performing an ANCOVA, having the pretest as a covariate.

 Author’s response:

 Thank you for your comment. Yes, we definitely agree that it is important to consider baseline values, especially for sleep data that present a lot of individual variability. In the linear mixed model, baseline values are also well accounted for (doi:10.1016/S0076-6879(04)84010-2). In this article, Van Dongen et al states that ANCOVA could be performed for this kind of data, but ANCOVA is a fixed-effects method and is more restrictive than mixed effects regression analysis. To be even more accurate in the results of the linear mixed model, we have added the 95% confidence interval, as did this study for example (Table 3, doi: 10.3390/ijerph18157999).

The unit should be added to Figure 2.

Author’s response: units were added to Figure 2.

-Results section: could you further interpret the results?

Author’s response: We have reviewed all our result section. We made the results more detailed. We hope you will find at your convenience.

Specific point

-Line 131: Please include the Epworth scale with the other subjective sleep measurements.

Author’s response: We included the Epworth scale with the other subjective sleep measurements.

-Line 135-140: please built upon this paragraph. Further information on the analysis of core body temperature by the e-celsius bodycap should be added. For instance how the Batyphase (°C), Acrophase (°C), and Amplitude (°C) were calculated?

Author’s response: We modified this requested paragraph as requested. Because the temperature data were accurate to 0.0001, we were able to accurately determine the exact timing (to within a 30 seconds range) and value over a 24-hour period of the minimum temperature (batyphase), maximum temperature (acrophase), and amplitude (acrophase value - batyphase value). In our study, we used core temperature as a reflection of the autonomic nervous system and not as a reflection of the circadian rhythm. Thus, only batyphase and acrophase values, not their timing, were reported.

The "e-celsius bodycap" capsule was used to measure the patient's core temperature continuously for at least 24 hours. Although no individual calibration of pills was performed, previous research has shown the equipment to be highly accurate and reliable(doi: 10.1249/MSS.0000000000001403). The time of intake was standardized for all participants at 3 pm. Participants should not perform any intense exercise at least 4 hours before bedtime. The temperature was recorded at 30 s intervals. The data were analyzed by the e performance manager software. The values of batyphase (minimum temperature) and acrophase (maximum temperature) over a 24-hour period (3pm-3pm), the amplitude (temperature difference between acrophase and batyphase) and the average temperature over the time spent in bed were recovered for analysis.”

-The title of table 2 is not appropriate

Author’s response: We changed the title of Table 2.

- The conclusions must be humble, based on the sample size, with suggestions.

Author’s response: We rephrased our conclusion.

Our results suggest that exercise interventions should be built to induce a rise in body temperature in order to induce sleep-promoting adaptations and rebounds. Further studies with larger sample sizes are needed to confirm these results.”

Round 2

Reviewer 1 Report

The authors have done a good job addressing my comments.

Author Response

Thank you

Reviewer 2 Report

Dear authors,

I have some more comments about your manuscript:

1. update aerobic exercise in the title and the abstract. I agree with you about aerobics; I was mistaken in writing about anaerobic previously.

2. you still continued using insomnia and sleep interchangeably. You must be more consistent with the terms: if your paper is focused on insomnia, you should use insomnia rather than sleep

3. It is unclear which is the gap in the preexisting literature and how your paper covers it. What does your paper add to the literature?

4. how did you evaluate the moderator effect (study aim)? I cannot see a moderator effect analysis. In my opinion, the linear mixed model cannot identify a moderator effect.

5. how and where did you comment on the correlation analysis?

6. along with citations 26 and 27, I suggest adding doi: 10.1038/s41598-022-25014-7 and DOI: 10.5603/GP.a2020.0172.

Author Response

  1. update aerobic exercise in the title and the abstract. I agree with you about aerobics; I was mistaken in writing about anaerobic previously.

Author’s response: We took your comments into account and modified the title and the abstract as you suggested (page 1, line 2 and 18)

  1. you still continued using insomnia and sleep interchangeably. You must be more consistent with the terms: if your paper is focused on insomnia, you should use insomnia rather than sleep

Author’s response: Thank you for your relevant comment. We have modified in line 48,50,56,61,66,127,279,285,306,316,317,334.

  1. It is unclear which is the gap in the preexisting literature and how your paper covers it. What does your paper add to the literature?

Author’s response: In the literature, the effect of a physical activity program on insomnia symptoms is well documented, and articles point to a decrease in insomnia symptoms through exercise. However, the results on the effect of a physical activity program on objective sleep parameters in insomniacs are more equivocal. Some show improvement in objective sleep parameters and others do not. Thus, our study brings new results on this debate.

Also, the optimal physical activity modalities to improve sleep are not known. Articles in the literature use different exercise modalities. Thus, by measuring core temperature in parallel with insomnia measurements, the idea was to identify whether physical activity by its effect on nocturnal core temperature could be related to insomnia symptoms. Was the evolution of one related to the other? Knowledge of these interaction mechanisms could allow the optimization of PA programs by inducing identified and specific adaptations.

  1. how did you evaluate the moderator effect (study aim)? I cannot see a moderator effect analysis. In my opinion, the linear mixed model cannot identify a moderator effect.

Author’s response: Indeed, we have replaced “moderators effect” by “interaction mechanisms” (Line 66, 315)

  1. how and where did you comment on the correlation analysis?

Author’s response: We explained the correlations as an association between the 2 variables. In no case did we talk about a cause and effect relationship.

The correlation between epworth and ISI was discussed page 9, line 306-309.

Correlations between core temperature parameters and ISI were discussed. Page 9, line 344-347.

  1. along with citations 26 and 27, I suggest adding doi: 10.1038/s41598-022-25014-7 and DOI: 10.5603/GP.a2020.0172.

Author’s response: This article (DOI: 10.5603/GP.a2020.0172) was already present (citation 26). We have added the other article, as you suggested (citation 28).

Reviewer 3 Report

No further comments

Author Response

Thank you

Round 3

Reviewer 2 Report

I suggest the authors add or stress the following point in the manuscript (reply 3):

Author’s response: In the literature, the effect of a physical activity program on insomnia symptoms is well documented, and articles point to a decrease in insomnia symptoms through exercise. However, the results on the effect of a physical activity program on objective sleep parameters in insomniacs are more equivocal. Some show improvement in objective sleep parameters and others do not. Thus, our study brings new results on this debate.

Also, the optimal physical activity modalities to improve sleep are not known. Articles in the literature use different exercise modalities. Thus, by measuring core temperature in parallel with insomnia measurements, the idea was to identify whether physical activity by its effect on nocturnal core temperature could be related to insomnia symptoms. Was the evolution of one related to the other? Knowledge of these interaction mechanisms could allow the optimization of PA programs by inducing identified and specific adaptations.

Author Response

Author’s response: We took your comments into account and added the following sentences at the end of introduction.

"The first aim of this study was to investigate the effect of a 12-week aerobic exercise training intervention on insomnia severity and objective sleep in adult women suffering from insomnia. Indeed, the effect of a physical activity program on insomnia symptoms is well documented, and articles point to a decrease in insomnia symptoms through exercise. However, the results on the effect of a physical activity program on objective sleep parameters in insomniacs are less conclusive. Thus, our study brings new results on this debate. The second objective was to identify some potential interaction mechanisms of the association between insomnia and exercise with a special focus on the core temperature. Knowledge of these interaction mechanisms could help to identify the optimal exercise modalities to reduce insomnia and allow the optimization of PA programs by inducing identified and specific adaptations."